# Get Fooled for the Right Reason: Improving Adversarial Robustness through a Teacher-guided Curriculum Learning Approach

**Anindya Sarkar** *
Indian Institute of Technology, Hyderabad
`anindyasarkar.ece@gmail.com`

**Anirban Sarkar** *
Indian Institute of Technology, Hyderabad
`cs16resch11006@iith.ac.in`

**Sowrya Gali** *
Indian Institute of Technology, Hyderabad
`cs18btech11012@iith.ac.in`

**Vineeth N Balasubramanian**
Indian Institute of Technology, Hyderabad
`vineethnb@iith.ac.in`

## Abstract

Current SOTA adversarially robust models are mostly based on adversarial training (AT) and differ only by some regularizers either at inner maximization or outer minimization steps. Being repetitive in nature during the inner maximization step, they take a huge time to train. We propose a non-iterative method that enforces the following ideas during training. Attribution maps are more aligned to the actual object in the image for adversarially robust models compared to naturally trained models. Also, the allowed set of pixels to perturb an image (that changes model decision) should be restricted to the object pixels only, which reduces the attack strength by limiting the attack space. Our method achieves significant performance gains with a little extra effort (10-20%) over existing AT models and outperforms all other methods in terms of adversarial as well as natural accuracy. We have performed extensive experimentation with CIFAR-10, CIFAR-100, and TinyImageNet datasets and reported results against many popular strong adversarial attacks to prove the effectiveness of our method.

## 1 Introduction

Ever since deep neural network (DNN) models have emerged as the de facto technique to be applied for many vision problems, adversarial robustness has emerged as a critical need. Goodfellow et al. [1] identified this serious issue to show very different predictive behavior of a DNN model with similar looking images. Many efforts have followed since either to come up with fooling techniques [1, 2, 3, 4, 5, 6] or defend deep models against them [3, 7, 8, 9, 10, 11, 12, 13, 14, 15, 16, 17, 18, 19, 20, 21, 22, 23, 24, 25, 26]. Nevertheless, both these research directions are important at this moment and require more attention. In this work, our effort lies on devising an adversarially robust training technique.

Adversarial training (AT) [3] is the most widely used method for adversarial robustness and most of the improvements have since come by adding regularizers without changing the min-max formulation. The regularizers are added either in the inner maximization [27, 8, 15] or outer minimization [19, 14] term. Though AT-based methods are shown to perform well, they incur additional cost due to an iterative inner maximization step. Our proposed robust training method circumvents this fundamental issue.

With an objective of not following the costly min-max optimization of AT, we revisit adversarial robustness in terms of standard model training. Adversarially robust models are shown to satisfy

---

*equal contribution

better alignment between saliency and object features in [28]. We enforce such alignment through training to achieve model robustness. This is accomplished by forcing saliency of the main model to follow the object features, provided by saliency of a pre-trained reference model. We hypothesize that perturbing only the most discriminative part of an image should trigger a model to change its decision about that image. In other words, perturbing other non-important pixels shouldn't affect model decision, and such a model can achieve adversarial robustness. This approach is achieved by progressively narrowing the object-discriminative region according to its importance in a curriculum learning sense and forcing an adversary to perturb only those pixels while changing model's decision. A model trained with this approach restricts the perturbation only to the object pixels when attacked. This diminishes the range of possible perturbations and limits the attack strength, reducing the chance of the model to change its decision for the perturbed image. Being a non-iterative method, our approach not only takes considerably less time for training compared to AT, but also outperforms iterative methods (such as [3, 7, 8, 27, 16, 13, 14, 11, 12, 17, 15]) and non-iterative methods (such as [29, 10, 9, 18]) significantly in both natural and adversarial accuracies.

Our key contributions are summarized as below.

- We propose a non-iterative novel robust training method, which outperforms recently proposed SOTA techniques, irrespective of their type being iterative or non-iterative, in terms of both natural and adversarial accuracies against a wide range of attacks.
- Being a non-iterative method makes it easily applicable to any large dataset to achieve an adversarially robust model. Our method takes 10-20% of time compared to any adversarial training technique.
- Our method attains a clean accuracy which is much closer to the performance of naturally trained models compared to other robust models.
- We perform extensive experimentation on CIFAR-10, CIFAR-100, TinyImageNet datasets and report comparative results against all other, including recently proposed, adversarial robustness techniques. We also present various studies in detail to analyze the effectiveness of our method.

## 2  Background

**Adversarial robustness**: Naturally trained deep models are shown to be fooled easily by image perturbations which are human-imperceptible [1]. Extra measures need to be taken to make them adversarially robust when they stick to their decision even an image is perturbed. Consider an image classifier $F(x; \theta) : x \longrightarrow \mathbb{R}^c$ with parameters $\theta$, that maps input image $x$ to a $c$-dimensional output. The network $f$ is called adversarially robust if:

$$\underset{i \in c}{\mathrm{argmax}} f_i(x; \theta) = \underset{i \in c}{\mathrm{argmax}} f_i(x + \delta; \theta) \tag{1}$$

where $\delta \in B(\epsilon)$ i.e. $\delta : ||\delta||_p \leq \epsilon$.

**Adversarial training (AT)**: AT [3] was proposed for achieving robustness against adversarial examples, which is represented as the below loss function:

$$L(x, y) = \mathbb{E}_{(x,y) \ D} \big[ \max_{\delta \in B(\epsilon)} L(x + \delta, y) \big] \tag{2}$$

$\delta$ is found by projected gradient descent (PGD) which is given by the iterative gradient step as below:

$$\delta \longleftarrow proj[\delta - \eta sign(\nabla_\delta L(x + \delta, y))] \tag{3}$$

where $Proj(x) = \underset{e \in B(\epsilon)}{\mathrm{argmin}} ||x - e||$. Effectively, AT is comprised of an inner maximization which generates a perturbed image within an $\epsilon$ ball and the outer minimization tunes the model parameters based on the perturbed images.

## 3  Related Work

**Robust model training.** Many efforts on making DNN models adversarially robust have been proposed in the last few years and shown to perform well against adversarial attacks. [27] proposed TRADES, which perturbs an image by maximizing the $L_2$ distance of the logits and used that as a regularizer to smoothen the decision boundary. *Misclassification Aware adveRsarial Training* (MART) [14], on the other hand, treats the misclassified examples differently and adds a regularizer term to weight them by the *misclassification score* for better robustness. Recently, [13] found a strong positive correlation between robust generalization gap and flatness of weight loss landscape. This motivated an *Adversarial Weight Perturbation* (AWP) mechanism which adversarially perturbs

both inputs and weights in the adversarial training framework to regularize the flatness of weight loss landscape. Zhang et al proposed *Geometry aware Instance Reweighted adversarial training* (GAIRAT) [16] method, which assigns larger weights on the losses of adversarial data, whose natural counterparts are closer to the decision boundary. Contrastively, GAIRAT gives smaller weights on the losses of adversarial data, whose natural counterparts are further away from the decision boundary.

**Curriculum learning-based adversarial training.** A few recent works have modified the adversarial training strategy incorporating a curriculum learning strategy. *Curriculum Adversarial Training* (CAT) [11] proposed a curriculum-based adversarial training strategy by progressively increasing the number of PGD steps during training. [12] proposed a dynamic training strategy to gradually increase the convergence quality of the generated adversarial examples, and also a method to quantitatively evaluate the convergence quality of adversarial examples motivated by the Frank-Wolfe optimality gap. [17] tried to employ the fact that, during initial phase of training, fitting with most difficult adversarial data makes the learning extremely hard for DNNs. They hence proposed *Friendly Adversarial Training* (FAT), adhering to the spirit of curriculum learning, which learns initially from the least adversarial and progressively utilizes increasingly more adversarial data. Another recent method *Adversarial Training with Early Stopping* (ATES) [15] proposed curriculum loss as the inner maximization step which depends on a difficulty parameter that are gradually increased as the training progresses. Our method is very different from these methods on how the curriculum learning strategy is employed.

**Non-iterative adversarial robustness training.** There are very few efforts in devising a robust model through a non-iterative approach that differs from AT [3]. [30] showed that models trained using single-step adversarial training methods are susceptible to multi-step white box attacks, such as PGD [3]. Recently, [22] proposed a single-step adversarial training method using Dropout Scheduling, which improves adversarial robustness against multi-step white box attacks and achieves par results compared to [3]. But all these single-step adversarial robustness methods still lag far behind current state-of-the-art (SOTA) methods that use the Adversarial Training framework. In terms of methods that use saliency, *Jacobian Adversarially Regularized Network* (JARN) [10] improves model robustness by matching the gradient of loss w.r.t. the attacked image to the actual image. A very similar method was proposed by [9] which employs a discriminator to compare between the Jacobian and the image saliency, while JARN [10] compares the image to the transformed version of the Jacobian through an adaptive network. Our method differs from these as we enforce localization of saliency (w.r.t true class score) and restrict the perturbation set to pixels of the most discriminative part of the object. We explain our proposed method in detail below.

## 4    Methodology

Adversarial robustness targets to improve model robustness against adversarially perturbed images. Also for adversarially robust models, attribution maps tend to align more to actual image compared to naturally trained models. We study this connection below.

**Robustness and alignment**: For an n-class classifier $F(x) = \underset{i}{\mathrm{argmax}}\,\Psi^i(x)$ where $\Psi = (\Psi^1, ..., \Psi^n) : X \longrightarrow \mathbb{R}^n$ be differentiable in $x$. Then we call $\nabla\Psi^{F(x)}$ the saliency map of $F$ and the alignment [28] with respect to $\Psi$ in $x$ is represented by

$$\alpha(x) := \frac{|\langle x, \nabla\Psi^{F(x)}(x)\rangle|}{||\nabla\Psi^{F(x)}(x)||} \tag{4}$$

Connection of robustness with alignment was studied in [28]. This connection is specifically formalized in Theorem 2, which states that a network's linearized robustness ($\hat{\rho}$) around an input $x$ is upper bounded by the binarized alignment term $\alpha^+$ as:

$$\hat{\rho}(x) \leq \alpha^+(x) + \frac{C}{||g||} \tag{5}$$

Here $C$ is a constant, the linearized robustness $\hat{\rho}(x)$ is given by

$$\hat{\rho}(x) := \min_{j \neq i^*} \frac{\Psi^{i*}(x) - \Psi^j(x)}{||\nabla\Psi^{i*}(x) - \nabla\Psi^j(x)||} \tag{6}$$

Also, $g$ is the Jacobian of the top two logits i.e. $g = \nabla(\Psi^{i*}(x) - \Psi^{j*}(x))$ and binarized alignment i.e. $\alpha^+$ is given by

$$\alpha^+(x) = \frac{|\langle x, \nabla(\Psi^{i*} - \Psi^{j*})(x)\rangle|}{||\nabla(\Psi^{i*} - \Psi^{j*})(x)||} \tag{7}$$

Here $j^*$ is the minimizer of Eqn 6. We also have $\alpha(x) = \alpha^+(x)$ for linear model and binary classifier. Eqn 5 explains the deviation of different terms for linearized robustness in case of neural network. Also, a small error term in Eqn 5 implies that robust networks yield better alignment i.e. more interpretable saliency maps.

Adversarial robustness can also be viewed from an angle where an image can be perturbed, to change the model decision, only through perturbing the pixels of the object and not through any pixel outside of the object in the image. We incorporate these ideas through our two-phase training method to achieve an adversarially robust model. Current methods for adversarial robustness in literature have two key drawbacks: (a) Adversarially robust models are shown to perform poorly on clean data. There exists a clear trade-off between adversarial and natural accuracies which is explored in [31]. (b) Almost all SOTA adversarial robustness methods rely on iterative adversarial training frameworks, which makes them very costly to apply. In the proposed method, we aim to solve both these issues, as the model, trained by our method, pushes the bar of both adversarial accuracy and natural accuracy. At the same time, our training strategy does not rely on the iterative adversarial training framework, and thus is very fast.

### 4.1 Teacher-guided Saliency-based Robust Training

**First Phase - Enforcing Alignment.** This phase incorporates the alignment of the attribution map to the object in the image, as explained before. Let's assume, we have a pre-trained Teacher network (represented as $f_T$). We also consider a student network, represented by a neural network $f_S$, parameterized by $\theta$ and a discriminator network $f_{disc}$ parameterized by $\phi$. Given an input image $x$, we obtain the saliency map from a pre-trained Teacher Network, which is denoted as $J_T^{TCI}$ (TCI represents true class index). Now, we maximize the true class prediction score of student network w.r.t input pixels and measure the net change in input pixels, which is represented as $J_S^{TCI}$.

Now for an image of dimension $h \times w$ with $c$ channels and $d = h \times w \times c$, $J_T^{TCI}$ can be considered as per-pixel gradient and represented as:

$$J_T^{TCI}(x) = \nabla \Psi^{f_T(x)} = [\nabla \Psi^{f_T(x_1)} ... \nabla \Psi^{f_T(x_d)}] \tag{8}$$

Similarly, $J_S^{TCI}$ is represented as:

$$J_S^{TCI}(x) = \nabla \Psi^{f_S(x)} = [\nabla \Psi^{f_S(x_1)} ... \nabla \Psi^{f_S(x_d)}] \tag{9}$$

We visualize the training process in first phase of figure 1. Here we are trying to enforce the fact that the set of pixels, responsible to increase the true class prediction score, are same as actual object pixels which are highlighted by the reference teacher saliency map. This implies imposing similarity between the two saliency maps $J_T^{TCI}$ and $J_S^{TCI}$. Inspired from [9], here we consider the concept of discriminator from GAN[32], which tries to differentiate between real and fake images and backpropagate a signal that forces the model to generate realistic looking images. Here, our sole purpose of using a discriminator is to influence our model to generate better saliency that matches the teacher saliency. Hence, we minimize the following objective function:

$$\theta_{optimum} = \operatorname*{argmin}_{\theta}[\mathcal{L}_{CE} + \beta \mathcal{L}_{Robust}]; \tag{10}$$

Where cross-entropy loss $\mathcal{L}_{CE} = \mathbb{E}_{(x,y)}[-y^T \log f_S(x)]$ and $\mathcal{L}_{Robust}$ is defined as below:

$$\mathcal{L}_{Robust} = \mathbb{E}_{J_T}[\log f_{disc}(J_T^{TCI})] + \mathbb{E}_{J_S^{TCI}}[\log(1 - f_{disc}(J_S^{TCI}))] \tag{11}$$

Now it can be shown that the global minimum of $\mathcal{L}_{Robust}$ is achieved when $J_S^{TCI}$ matches $J_T^{TCI}$ [10], which justifies using the discriminator loss for our purpose. We also add a loss term to minimize the $l_2$ distance between $J_T^{TCI}$ and $J_S^{TCI}$, which is represented as $L_{diff}$ and defined as below:

$$\mathcal{L}_{diff} = ||J_S^{TCI} - J_T^{TCI}||_2^2 \tag{12}$$

Hence, we minimize the complete objective function as below:

$$\theta_{optimum} = \operatorname*{argmin}_{\theta}[\mathcal{L}_{CE} + \underbrace{(\beta \mathcal{L}_{Robust} + \gamma \mathcal{L}_{diff})}_{alignment\ loss}]; \tag{13}$$

At the same time, The discriminator network is trained as follows:

$$\phi_{optimum} = \operatorname*{argmax}_{\phi}[\mathcal{L}_{Robust}] \tag{14}$$

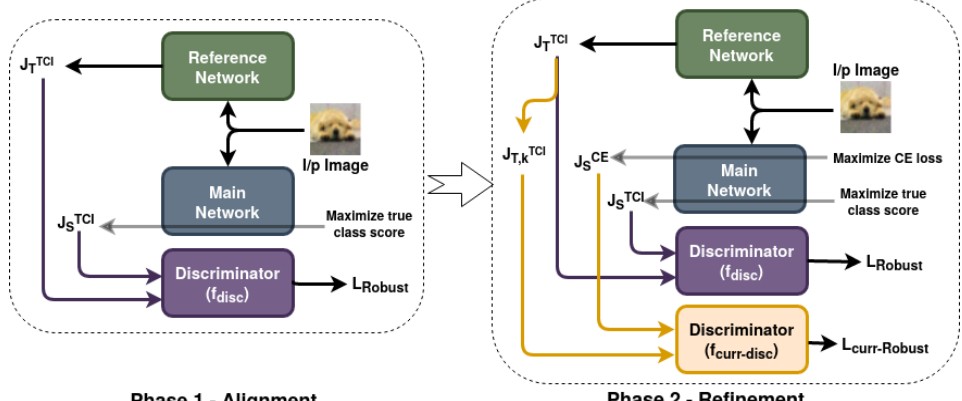

**Phase 1 - Alignment**         **Phase 2 - Refinement**

Figure 1: Our proposed training strategy. Phase 1 enforces alignment and phase 2 enforces curriculum style training by reducing $k$ in $J_{T,k}^{TCI}$ at every step.

The model achieved through this phase of training can generate saliency maps that can localize the object nicely. This model itself can perform on par with AT[3] in terms of both adversarial and natural accuracy, and are shown in 6. With the target of attaining better performance, we continue with the second phase of training which is explained below.

**Second Phase - Model Refinement.** In this phase, we want to ensure that the decision of the model can be changed only by perturbing the object pixels. Here we bring curriculum style learning in picture by gradually shortening the set of pixels which are allowed to perturb in order to reduce true class prediction score. The set of pixels are selected based on discriminativeness of the object parts and which is decided by the teacher saliency. At every step of this phase, the training enforces that the attacker has to change the image by only perturbing among some fixed amount of top pixels from the whole object i.e. the highlighted part of the saliency map given by the teacher. From another perspective, if an attacker, in order to change the model's decision, has only very few options to perturb object pixels compared to all the pixels in the input image, which drastically reduces the adversarial attack effect on input image. Hence this is implied that the search space of pixels will be very limited during each iteration of adversarial attack step.

Now, during first step of this phase of training, top 90% of the pixels from the teacher saliency are considered i.e. the allowed set of pixels for modification by the student model is reduced to that top 90%. Hence, the training will enforce that only these set of pixels are responsible to maximize the loss i.e. decrease the true class prediction score. Here we follow a curriculum style training and after training the first step for few epochs, we consider only top 80% of the pixels from the teacher saliency in the second step. We continue in this fashion with training every step for few epochs (predecided and kept same for every step) and stop after training with top 50% of teacher saliency. We explain the training for one step below with $k$ as the top percentage of pixels from teacher saliency.

As stated in the first phase of training, we obtain $J_T^{TCI}$ in the exact same way. Then, we keep top $k\%$ salient pixels of $J_T^{TCI}$, which is denoted by $J_{T,k}^{TCI}$, and make remaining less salient pixels of actual object zero. We obtain $J_S^{CE}$ by maximizing the CE loss of student network w.r.t. input pixels and then considering the net change in input pixels. Another discriminator network $f_{curr-disc}$ is considered which is parameterized by $\xi$.

Now we have $\mathcal{L}_{CE} = \mathbb{E}_{(x,y)}[-y^T log f_S(x)]$. Also, $J_S^{CE}$ can be considered as per-pixel gradient and represented as

$$J_S^{CE} = \nabla_x \mathcal{L}_{CE} = [\frac{\delta \mathcal{L}_{CE}}{\delta x_1}...\frac{\delta \mathcal{L}_{CE}}{\delta x_d}] \tag{15}$$

Now, following the similar motive in the first phase, our training objective for this phase is presented as follows:

$$\theta_{optimum} = \underset{\theta}{argmin}[\mathcal{L}_{CE} + (\underbrace{\beta \mathcal{L}_{Robust} + \gamma \mathcal{L}_{diff}}_{\text{alignment loss}} + \underbrace{\beta \mathcal{L}_{curr-Robust} + \gamma \mathcal{L}_{curr-diff}}_{\text{curriculum loss}})]; \tag{16}$$

where *alignment loss* is similar as given in eq.13, and $\mathcal{L}_{curr-Robust}, L_{curr-diff}$ are defined as below:

| Type | Curriculum | Methods | Clean | FGSM | PGD-5 | PGD-10 | PGD-20 | C&W | AA |
|---|---|---|---|---|---|---|---|---|---|
| Iterative Methods | NO | AT(PGD-7)[3] | 87.25 | 56.22 | 55.50 | 47.30 | 45.90 | 46.80 | 44.04 |
| | | FNT[7] | 87.31 | NA | NA | 46.99 | 46.65 | NA | NA |
| | | LAT[8] | 87.80 | NA | NA | 53.84 | 53.71 | NA | 49.12 |
| | | TRADES[27]* | 84.92 | 61.06 | NA | NA | 56.61 | 51.98 | 53.08 |
| | | GAIRAT[16] | 85.75 | NA | NA | NA | 57.81 | NA | NA |
| | | AWP-AT[13]* | 85.57 | 62.90 | NA | NA | 58.14 | 55.96 | 54.04 |
| | | MART[14]* | 84.17 | 67.51 | NA | NA | 58.56 | 54.58 | NA |
| | YES | CAT18[11] | 77.43 | 57.17 | NA | NA | 46.06 | 42.28 | NA |
| | | Dynamic AT[12] | 85.03 | 63.53 | NA | NA | 48.70 | 47.27 | NA |
| | | FAT[17] | 87.00 | 65.94 | NA | NA | 49.86 | 48.65 | 53.51 |
| | | ATES[15]* | 86.84 | NA | NA | NA | 55.06 | NA | 50.72 |
| Non-Iterative Methods | NO | SADS[29]+ | 82.01 | 51.99 | NA | 45.66 | NA | NA | NA |
| | | JARN-AT1[10] | 84.80 | 67.20 | 50.00 | 27.60 | 15.50 | NA | 0.26 |
| | | IGAM[9] | 88.70 | 54.00 | 52.50 | 47.60 | 45.10 | NA | NA |
| | | AT-Free[18] | 85.96 | NA | NA | NA | 46.82 | 46.60 | 41.47 |
| | YES | **OURS** | **90.63** | **67.84** | **63.81** | **61.44** | **59.59** | **61.83** | **54.71** |

Table 1: Results of natural and adversarial accuracies of different adversarial robustness techniques against different attacks on CIFAR-10 dataset. Here $*$ and $+$ with a method represents that the corresponding results are with WRN34-10 and WRN28-10 networks respectively.

$$\mathcal{L}_{curr-Robust} = \mathbb{E}_{J_{T,k}^{TCI}}[\log f_{curr-disc}(J_{T,k}^{TCI})] + \mathbb{E}_{J_S^{CE}}[\log(1 - f_{curr-disc}(J_S^{CE}))] \qquad (17)$$

$$\mathcal{L}_{curr-diff} = ||J_S^{CE} - J_{T,k}^{TCI}||_2^2 \qquad (18)$$

Please note, we set $\beta$ and $\gamma$ coefficients with discriminator loss and $l_2$ loss respectively according to their importance. Similar to $\mathcal{L}_{Robust}$ in 11, we can show that the global minimum of $\mathcal{L}_{curr-Robust}$ is achieved when $J_S^{TCI} = J_{T,k}^{TCI}$ [10], which substantiates the use of discriminator loss. At this stage, apart from the discriminator training at eq.14, the other discriminator network ($f_{curr-disc}$) is trained as follows:

$$\xi_{optimum} = \underset{\xi}{\mathrm{argmax}}[\mathcal{L}_{curr-Robust}] \qquad (19)$$

The second phase of training strategy is visualized in fig.1. The curriculum is imparted by gradually pruning out the least attributed pixels thus retaining only top $k$. $k$ is reduced in uniform steps of 10 starting from 100. For example, initially it is trained with $J_{T,100}^{TCI}$ for, say, 10 epochs, and then with $J_{T,90}^{TCI}$, $J_{T,80}^{TCI}$ and so on. Going in this fashion, ultimately, the model will be trained to perturb the image by modifying only the most discriminative part of the object to lower the class confidence. We now justify the requirement of two phase training and necessity of curriculum style learning.

**Why do we need both $\mathcal{L}_{Robust}$ and $\mathcal{L}_{diff}$ for the alignment loss?** Here both the losses i.e. $\mathcal{L}_{Robust}$ and $\mathcal{L}_{diff}$ together help serve the overall objective i.e. to generate a better saliency map that matches student saliency. They achieve global minima individually when both saliency maps match. A theoretical justification of attaining global minimum for $\mathcal{L}_{Robust}$ is shown in Theorem 3.1 of [9]. Adding both losses helped get stronger signals and attain better performance compared to using any one of them, which is also supported by our experimental findings shown in Sec 6.

**Why $J_S^{TCI}$ is considered during first phase and including $J_S^{CE}$ in the second phase?** Our robust model training is motivated by alignment of saliency map with the object features. This idea helps the student model to learn better object localization through saliency thus improving robustness and should be maintained throughout the training. Once the model has decent knowledge of the object after alignment phase, we include the concept of $J_S^{CE}$ in the training. Incorporating this help the model to learn the allowed set of pixels, to be perturbed, to reduce the class score which further boosts robustness. If we were to start the curriculum style training from the beginning, it would have disrupted the model from acquiring knowledge about the whole object.

**Why curriculum learning?** By introducing curriculum style learning, we try to enforce that most of the pixels, which are allowed to change, should belong to the most discriminative parts of the object. Also fewer pixels should be considered from the lesser discriminative parts of the object. As the discriminativeness is decided by the teacher saliency, we consider lesser number of pixels which is top most $k\%$ important part of the object. The value of $k$ is reduced at every step of curriculum and the model is trained to learn about the degree of object discriminativeness gradually.

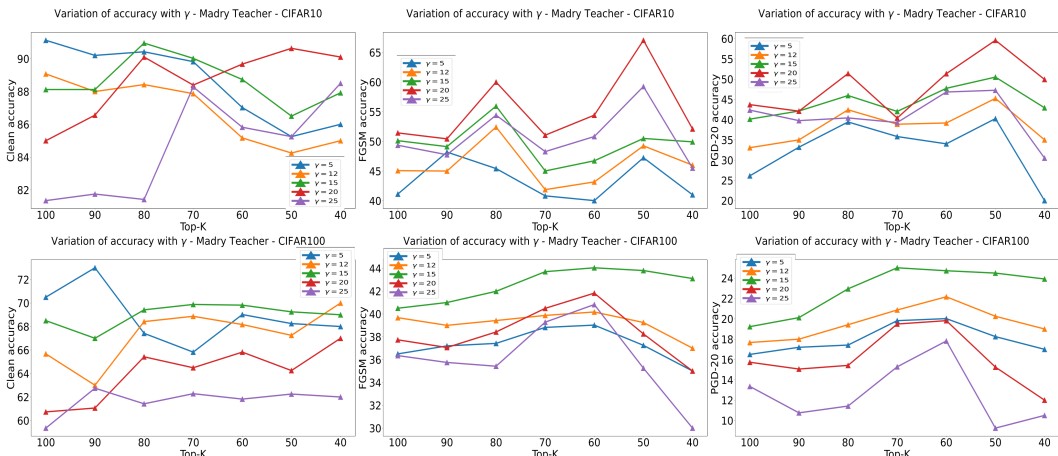

Figure 2: Effect of different $\gamma$ values and top-k% pixels during second phase of training on diffrent accuracies for CIFAR10 (top row) and CIFAR100 (bottom row). These plots are for clean, FGSM and PGD-20 accuracy after every $k$ for 5 different $\gamma$ values.

# 5 Experiments and Results

Here we explain different experimental settings and the evaluation results against various attacks to show the efficacy of our method on CIFAR10 and CIFAR100 datasets [33]. Our results on the TinyImageNet dataset is included in the Appendix due to space constraints.

**Experimental Settings.** For CIFAR10, we continue first phase of training for 100 epochs and 15 epochs

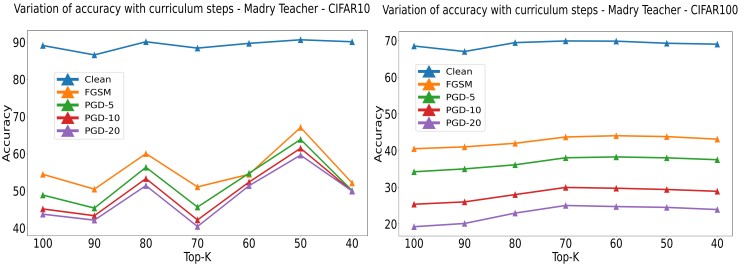

Figure 3: Effect of reducing top pixels during second phase of training. Considering adversarial accuracy for PGD-20 attack, best results are achieved after selecting top-50% and top-70% of teacher saliency for CIFAR10 and CIFAR100 respectively.

for each $k$ during second phase upto $k = 50$. While we keep $\beta = 2$ throughout the training, $\gamma$ is changed from 10 in first phase to 20 in second phase. Learning rate uniformly decays from 0.1 to 0.001 in first phase and from 0.001 to 0.0001 in second phase. For CIFAR100, we train with the same number of epochs as CIFAR-10 for both phases. While we keep $\beta = 2$ throughout the training, $\gamma$ is changed from 10 in first phase to 15 in second phase. Learning rate is also considered same as the training with CIFAR-10.

**Evaluation Against Adversarial Attacks.** Here we show evaluation results of our proposed model against various standard adversarial attacks [1, 3, 2] and recently proposed AutoAttack [5] , which is used as a strong attack against most of the recently proposed robust models. We consider many recently proposed methods [3, 7, 8, 27, 16, 13, 14, 11, 12, 17, 15, 29, 10, 9, 18] as baseline and report our performance on the same setting as their's on CIFAR-10 and CIFAR-100 datasets. For evaluation, we consider $L_\infty$ attack with $\epsilon = 8/255$ and use WRN32-10 as the main network to run all the experiments following [3] for both CIFAR10 and CIFAR100. While most of the methods, considered as baseline, follow same experimental setup and used WRN32-10 [34] architecture as their main network, some of them used a bigger WRN34-10 [34] network (which we point out in our result). We report these results in tables 1 and 2 for CIFAR-10 and CIFAR-100 datasets. These tables show that our method achieves best adversarial accuracy against all the attacks reported without sacrificing much on natural accuracy for both the datasets. We studied our model with random restarts, and observed variations of a negligible range ( +-0.02%) in our results compared to what we achieve without random restart. Our method thus also achieves state-of-the-art adversarial accuracy even when considering baselines with a random restart setup. It is worth mentioning that we surpass

| Training Type | Curriculum | Methods | Clean | FGSM | PGD-5 | PGD-10 | PGD-20 |
|---|---|---|---|---|---|---|---|
| Iterative Methods | NO | AT(PGD-7)[3] | 60.40 | 29.10 | 29.30 | 24.30 | 23.5 |
| | | FNT[7] | 60.27 | NA | NA | 22.44 | NA |
| | | LAT[8] | 60.94 | NA | NA | 27.03 | NA |
| | | TRADES[27]* | 58.55 | NA | NA | NA | 25.89 |
| | | IAAT [20] | 68.80 | NA | NA | 26.17 | NA |
| | YES | CAT18[11]* | 66.05 | NA | NA | NA | 10.91 |
| | | Dynamic AT[12]* | 54.71 | NA | NA | NA | 23.44 |
| | | ATES[15]* | 62.95 | NA | NA | NA | 28.05 |
| Non-Iterative Methods | NO | IGAM[9] | 62.39 | 34.31 | 29.59 | 24.05 | 21.74 |
| | | AT-Free[18] | 62.13 | NA | NA | NA | 25.88 |
| | YES | **OURS** | **69.88** | **43.71** | **38.03** | **30.04** | **28.32** |

Table 2: Results of natural and adversarial accuracies of different adversarial robustness techniques against different attacks on CIFAR-100 dataset. Here ∗ with a method represents that the corresponding results are with WRN34-10 network.

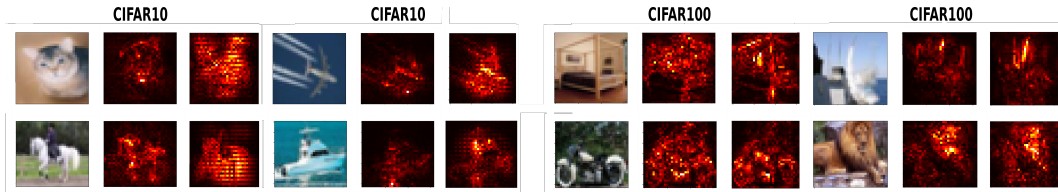

Figure 4: Visualizing effect of different teacher networks while training our student model. For each triplet of images above: *(Left)* Original image; *(Middle)* Saliency of our model trained with teacher 1 (adversarially trained on another dataset, finetuned on clean data of given dataset); *(Right)* Saliency of our model trained with teacher 2 (adversarially trained on given dataset)

all the other baseline methods including those using WRN34-10 for both CIFAR10 and CIFAR100. Please also note that the best model is achieved after training with $k = 50$ and $k = 70$ for CIFAR10 and CIFAR100 respectively. We experimented with increasing number of iterations of PGD attack on CIFAR-10 and achieved adversarial accuracy results as 59.59%, 58.50%, 58.22%, 58.04% for PGD-20, PGD-50, PGD-70, PGD-100 respectively; this shows that our method is fairly robust to increasing intensity of the PGD attack.

## 6 Discussion and Ablation Studies

**Effect of Curriculum Style Learning in Refinement Phase.**
As explained in sec.4, we reduce percentage of top discriminative pixels ($k$) from teacher saliency at every step and stop at top 50% of the pixels during refinement phase. We report adversarial and natural accuracies after every step with the best $\gamma$ values (20 for CIFAR10 and 15 for CIFAR100) in fig.3, which justifies the effectiveness of curriculum style learning.

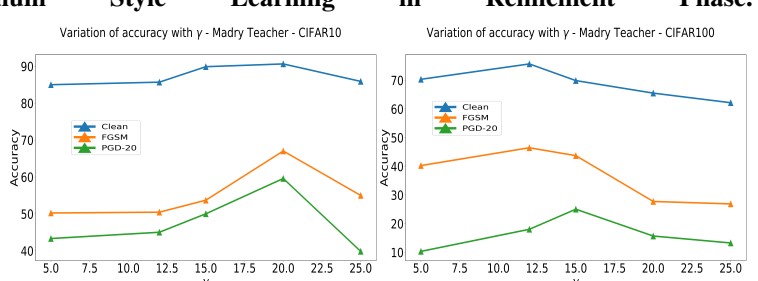

Figure 5: Effect of considering different values of $\gamma$ during second phase of training on final accuracies for CIFAR10 and CIFAR100

Our method achieves the best results for $k = 50$ and $k = 70$ for CIFAR10 and CIFAR100 datasets respectively (considering PGD-20 accuracy). We also provide detailed results for 5 different $\gamma$ values with different $k$ for clean, FGSM and PGD-20 accuracies for every $\gamma$. Fig.2 represents such result for CIFAR10 and CIFAR100. Please note that $k = 100$, for all the above mentioned figures, represents accuracy values by the model after first phase of training. We provide one more argument in support of curriculum learning for our method, where we experiment without curriculum learning and train refinement phase with the same loss function but with only a single $k$. Here we consider $k$ same as the best performing $k$ for training with curriculum learning (e.g. $k = 50$ for CIFAR10). Due to space constraints, we present details of this experiment with results in the Appendix.

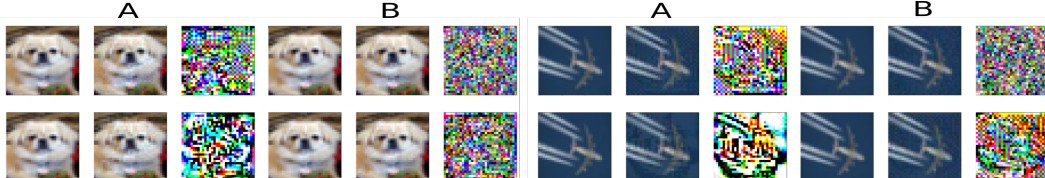

Figure 6: Effect of curriculum learning through visualizing perturbed images when the models are attacked by FGSM and PGD-20. For each group of 12 images, the top-left and bottom-left triplet corresponds to the actual image, perturbed image and net change (delta) in pixels when a model obtained after 1st training phase is attacked using FGSM and PGD-20 method respectively. Similarly, the top-right and bottom-right triplets corresponds to the same set of images obtained using a model obtained after 2nd phase of training. Here A and B represent images with models obtained after first and second phase respectively.

| Teacher | Nat. | FGSM | PGD5 | PGD10 | PGD20 |
|---------|------|------|------|-------|-------|
| A | 87.6 | 57.31 | 56.12 | 48.20 | 46.60 |
| B | **90.6** | **67.84** | **63.81** | **61.44** | **59.59** |

Table 3: Result on CIFAR-10

| Teacher | Nat. | FGSM | PGD5 | PGD10 | PGD20 |
|---------|------|------|------|-------|-------|
| C | **74.6** | **46.67** | 37.43 | 25.95 | 24.06 |
| D | 69.8 | 43.71 | **38.03** | **30.40** | **28.32** |

Table 4: Result on CIFAR-100

Robustness with Different teacher:(Note that all other hyperparameters kept same). A=Adversarially trained on TinyImageNet followed by finetuned on CIFAR10, B=Adversarially Trained on CIFAR10, C=Adversarially trained on CIFAR10 followed by finetuned on CIFAR100, D=Adversarially Trained on CIFAR100

**Effect of Adversarially Trained Teacher.** We consider a reference network which generates a nice saliency map that the main network would try to match. As we already discussed better alignment of saliency maps to the image features for adversarially robust models compared to naturally trained models, we considered two different models as reference networks for comaprison. For CIFAR10, one of the teachers is PGD-7 [3] trained model with TinyImageNet, followed by natural finetuned with CIFAR10, and the other one is PGD-7 [3] trained with CIFAR10. Similarly, for CIFAR100, we have results with teacher model which is PGD-7 [3] trained with CIFAR10 followed by naturally finetuned with CIFAR100 compared to PGD-7 [3] trained with CIFAR100. The results for CIFAR-10 and CIFAR-100 datasets are reported in tables 3 and 4, which depicts better accuracy numbers when our model is trained with full adversarially trained teacher. We also generate saliency maps with our model for both the teachers, which are presented in fig.4. As anticipated, we notice better saliency maps with full adversarially trained teacher for both CIFAR10 and CIFAR100.

**Importance of First Phase of Training.** This is an important question if we think of our gain by removing the first phase of training and using the AT model [3] directly, i.e. the teacher model for the second phase of training. As the training consists of cross-entropy as well as alignment losses, plugging the teacher model in place of the student model after the first phase ideally would incur zero alignment loss. But cross-entropy loss of such a model would be more compared to the model trained using our method. After the first phase, we achieve 1% to 2% improvement in natural as well as adversarial accuracies using our model compared to the teacher model as shown in Tables 11 and 12 in the Appendix. For comparing with AT, we report each loss term after phase 1 in Table 7, where the teacher model is of the similar architecture as the student i.e. WRN32-10. Moreover, in order to show the importance of the first phase training, we train the second phase, starting from AT model, instead of the model achieved after the first phase. Here the first phase of training sets the stage for curriculum style learning in the second phase, which is supported by our experimental findings suggesting that robustness of the final model (achieved after second phase of training) is much lesser if we use the AT model directly. Experimental results on CIFAR-10 are presented in Table 8.

**Effect of $\gamma$ Regularizer Coefficient.** We experimented with different $\gamma$ values during our training and reported the variation of accuracies for different $\gamma$s in fig.5. Please note that $\beta = 2$ and $\gamma = 10$ were kept always same for first phase of training, and different $\gamma$ values were tried during second

| Align Loss | Nat. | FGSM | PGD5 | PGD10 | PGD20 |
|------------|------|------|------|-------|-------|
| Remove | 88.9 | 59.98 | 59.02 | 56.90 | 55.84 |
| Keep | **90.6** | **67.84** | **63.81** | **61.44** | **59.59** |

Table 5: Results on CIFAR-10.

| Align Loss | Nat. | FGSM | PGD5 | PGD10 | PGD20 |
|------------|------|------|------|-------|-------|
| Remove | 67.3 | 39.32 | 34.67 | 27.14 | 24.45 |
| Keep | **69.8** | **43.71** | **38.03** | **30.04** | **28.32** |

Table 6: Results on CIFAR-100.

Results with keeping removing the alignment losses i.e. $L_{Robust}$ and $L_{diff}$ during 2nd training phase.

| Model | CE loss | Alignment loss |
|---|---|---|
| Our Model | 0.3594 | 0.009 |
| AT Model | 0.4956 | 0.0 |

Table 7: Cross-entropy and alignment loss values by our model and AT model [3] i.e. the teacher model after first phase of training for CIFAR-10.

| Model | Nat. | FGSM | PGD-20 |
|---|---|---|---|
| P1:OUR $\Longrightarrow$ P2:OUR | **90.63** | **67.84** | **59.59** |
| P1:AT $\Longrightarrow$ P2:OUR | 84.47 | 57.19 | 51.10 |

Table 8: Effect of starting the second phase training with the model achieved by our model after first phase training and AT model [3] for CIFAR-10.

| Combination | Natural | PGD20 |
|---|---|---|
| $\beta = 2$ and $\gamma = 20$ | **90.63** | **59.59** |
| $\beta = 0$ and $\gamma = 20$ | 87.08 | 54.83 |
| $\beta = 2$ and $\gamma = 0$ | 84.79 | 41.43 |

Table 9: Relative effect of $\beta$ and $\gamma$ on Natural and adversarial robustness performance for CIFAR-10.

phase of training only. During the second phase, we achieved best results with $\gamma = 20$ and $\gamma = 15$ for CIFAR10 and CIFAR100 respectively and fig.5 shows variation of $\gamma$ values during second phase only. Selecting higher $\gamma$ values compared to $\beta$ was required to enforce importance towards alignment and curriculum style training, which is also justified by our reported results.

**Relative Effect of $\beta$ and $\gamma$ Regularizer Coefficients.** We argued the importance of both losses i.e. $\mathcal{L}_{diff}$ and $\mathcal{L}_{Robust}$ in achieving our objective optimally in Sec 4. We support this claim experimentally by studying our model performance considering $\beta = 0$ and $\gamma = 0$ in Table 9 for CIFAR-10. The best result is attained with both $\beta$ and $\gamma$ compared to removing any one of them.

**Effect of Continuing Alignment Losses During Refinement Phase.** We start our model training with alignment based losses i.e. $L_{Robust}$ and $L_{diff}$ during first phase of training. Discontinuing them during second phase forces the model to completely focus on restricting allowed pixels for perturbation only and gradually forget knowledge about the object as a whole given by the teacher saliency. We justify this argument by reporting the final natural and adversarial accuracies with keeping and removing both the alignment based losses during second phase of training. We got better result by keeping both of them in the second phase of training which is reported in tables 5 and 6 for CIFAR-10 and CIFAR-100 respectively.

**Effect of Curriculum Learning in Restricting the Perturbation Set.** We visualize the effect of curriculum style learning by considering models before and after second phase of training and generating adversarially perturbed images by both of them. This is presented in fig.6 and these models are termed as A and B respectively. For better clarification, we take the difference between original and perturbed images and call it delta, which is an indication of all the perturbations. Please note that the pixels of delta are normalized between 0 and 1, which represents the maximum possible perturbation for a pixel as a white pixel for example. In case of B model, clearly the perturbations are much lesser than A model for both FGSM and PGD-20 attack for both the example images. As the model is trained with enforcing restriction of perturbation during the second phase of training, these visualizations are justified. We provide more such visualizations in the Appendix section.

## 7 Conclusion

In this work, we propose a new non-iterative method to achieve adversarial robustness that operates at 10-20% of the training cost of traditional adversarial training methods. Our method significantly outperforms state-of-the-art methods on adversarial accuracy without affecting natural accuracy, which demonstrates the efficiency and practical applicability of our training strategy.

**Broader Impact.** *Pros*: Methods like Adversarial Training take a tremendous amount of time and computational resources. Our method is very efficient in training time and produces SOTA results without compromising on natural accuracy. This allows our method to be practically useful for adversarial robustness efforts. *Cons*: Owing to the different Jacobians used, our method's efficacy in memory consumption may require improvement. There are no other known socially detrimental effects of this work.

**Acknowledgements and Funding Transparency Statement.** This work has been partly supported by the Govt of India UAY program, Honeywell and a Google Research Scholar Award.

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
