# Appendix

In this appendix, we provide details that could not be included in the main paper owing to space constraints, including: (i) evaluation results on TinyImageNet dataset against adversarial attack; (ii) robustness of our model after alignment phase; (iii) effect of curriculum learning in progressively reducing $k$ rather than training the second phase with a fixed $k$; (iv) more visualizations on effect of different teachers experiment continuing from Fig 4; as well as (v) more visualizations on effect of curriculum learning in restricting the perturbation set, continuing from Fig 6.

Please note that we used a standard computing server with 3 GeForce GTX 1080 Ti GPUs each of 12GB for all the experiments in our work. The total compute time for completion of training for our model is 6 hours (including phase 1 and 2), while the Adversarially Trained (AT) model takes 30 hours for CIFAR10 with WRN32-10 architecture (our method takes approx 20% of the training time of an AT model). Also for both CIFAR10 and CIFAR100, our model training consumed $\sim 11$GB of GPU with a batch size of 32; AT training takes $\sim 7$GB of GPU with the same batch size. For TinyImageNet, our model and the AT trained model consume similar amounts of memory i.e. $\sim 11$GB and $\sim 7$GB respectively with a batch size of 16.

## A Results on TinyImageNet

We begin with describing the experimental settings, followed by the results of our method against a PGD-20 adversarial attack on the TinyImageNet dataset.

**Experimental Settings.** We perform the first phase of training for 100 epochs, and subsequently train for 15 epochs for each $k$ during the second phase upto $k = 50$. While we keep $\beta = 2$ throughout the training, $\gamma$ is changed from 10 in first phase to 50 in second phase. Learning rate uniformly decays from 0.1 to 0.001 in first phase and from 0.001 to 0.0001 in second phase.

**Evaluation Against Adversarial Attacks.** Here we show evaluation results of our proposed model against the standard adversarial attack [3]. Results for TinyImageNet are available for very few methods such as [3, 19, 27, 9], and we considered them as our baselines. For evaluation, we consider $L_\infty$ attack with $\epsilon = 8/255$ and use WRN32-10 as the main network to run all the experiments following [3]. While one of the methods, considered as baseline, follow similar experimental setup and used WRN32-10 [34] architecture as their main network, others used a bigger WRN34-10 [34] network (which we point out in our result). We report these results in Table 10. The results show a similar trend as for CIFAR10 and CIFAR100 in the main paper. Our method achieves best adversarial accuracy against standard PGD-20 adversarial attack without better natural accuracy than the baselines. We surpass all the other baseline methods, including those using WRN34-10.

| Training Type | Curriculum | Methods | Clean | PGD-20 |
|---|---|---|---|---|
| Iterative Methods | $NO$ | AT [3]* | 30.65 | 6.81 |
| | | ALP[19]* | 30.51 | 8.01 |
| | | TRADES[27]* | 38.51 | 13.48 |
| | $YES$ | NA | NA | NA |
| Non-Iterative Methods | $NO$ | IGAM[9] | 54.26 | 10.12 |
| | $YES$ | **OURS** | **61.37** | **18.38** |

Table 10: Results of natural and adversarial accuracies of different adversarial robustness techniques against PGD-20 attacks on Tiny-Imagenet dataset. Here $*$ with a method represents that the corresponding results are with WRN34-10 network. We used WRN32-10 for our method.

## B Results of Our Model after Alignment Phase

In the main paper, we explained the importance of two-phase training in achieving a robust model and performed many experiments that validated the performance of the proposed method. While the second phase of training is shown to boost the robustness of the model, we observe that the first phase can by itself attain satisfactory robustness (better than standard adversarial training). We compare the results of our method after alignment phase with [3] in Tables 11 and 12 for CIFAR10 and CIFAR100 respectively, which support this claim and highlight the importance of the alignment phase for robust model training.

| Phase | Nat. | FGSM | PGD5 | PGD10 | PGD20 |
|-------|------|------|------|-------|-------|
| AT[3] | 87.2 | 56.22 | 55.50 | 47.30 | 45.90 |
| 1st | 88.7 | 56.71 | 55.12 | 49.68 | 47.03 |
| 2nd | **90.6** | **67.84** | **63.81** | **61.44** | **59.59** |

Table 11: Results on CIFAR-10.

| Phase | Nat. | FGSM | PGD5 | PGD10 | PGD20 |
|-------|------|------|------|-------|-------|
| AT[3] | 60.4 | 29.10 | 29.30 | 24.30 | 23.50 |
| 1st | 62.4 | 36.21 | 31.02 | 25.19 | 22.81 |
| 2nd | **69.8** | **43.71** | **38.03** | **30.04** | **28.32** |

Table 12: Results on CIFAR-100.

Comparative results of our model after first phase of training with AT[3]. Results with second phase of training are added for completeness.

| Method | Nat. | FGSM | PGD5 | PGD10 | PGD20 |
|--------|------|------|------|-------|-------|
| Direct | 89.5 | 64.77 | 61.29 | 59.52 | 57.93 |
| Curr | **90.6** | **67.84** | **63.81** | **61.44** | **59.59** |

Table 13: Results on CIFAR-10.

| Method | Nat. | FGSM | PGD5 | PGD10 | PGD20 |
|--------|------|------|------|-------|-------|
| Direct | 68.6 | 41.65 | 36.24 | 28.57 | 27.48 |
| Curr | **69.8** | **43.71** | **38.03** | **30.04** | **28.32** |

Table 14: Results on CIFAR-100.

Comparative results between curriculum style learning and directly training with a single $k$ during second phase of training

## C  Effect of Curriculum Learning on Progressively Reducing $k$

We performed several ablation studies in Sec 6 of the main paper to show the importance of curriculum learning during the refinement phase of training. Here, we conduct another experiment where the refinement phase training is carried out with only a single value of $k$. We choose $k$ to the value that showed the best performance in the curriculum learning approach for fair comparison. The results in Fig 3 show that the best performance is obtained with $k = 50$ and $k = 70$ for CIFAR10 and CIFAR100 respectively. We hence consider these values for $k$ in this experiment and report the results in Tables 13 and 14. Note that all other hyperparameters for this experiment with a single $k$ are kept same as that of curriculum-style learning. The results show that the refinement phase with a single $k$ shows promise by itself, but does not surpass the results accomplished by curriculum learning.

## D  More Visualizations on Effect of Adversarially Trained Teacher

In this section, we study the importance of the teacher model for our model training, since the main network attempts to match the saliency generated by the teacher. This was discussed in Section 6 with visualizations in Fig 4. Here, we provide more such visualizations in Fig 7 which we couldn't provide in the main paper due to space constraints.

We consider two different teachers for this experiment for both CIFAR10 and CIFAR100. For CI-FAR10, one of the teachers is a model adversarially trained using PGD-7 on TinyImageNet, followed by finetuning on clean CIFAR10 data, and the other teacher is a model adversarially trained using PGD-7 on CIFAR10. Similarly, for CIFAR100, one of the teachers is a model adversarially trained using PGD-7 on CIFAR10, followed by finetuning on clean CIFAR100 data, and the other teacher is a model adversarially trained using PGD-7 on CIFAR100. A teacher trained with full adversarial training on a given dataset should generate better saliency maps, and hence, the corresponding student model should learn better. For every image, Fig 7 shows saliency maps generated by student model trained with two different teachers, as mentioned in the same sequence above for CIFAR10 and CIFAR100. We added more such examples for images from CIFAR10 dataset in Fig 8 for better understanding. It is evident that a student trained with a teacher trained with full adversarial training generates better saliency maps, which translates to better adversarial robustness. These observations are aligned with the results in Tables 3 and 4 of the main paper. For better understanding of the effect of two different teachers, we added more examples

### D.1  More Visualizations on Effect of Curriculum Learning in Restricting Perturbation Set

We generate adversarially perturbed images from both models after alignment and refinement phases. Then calculating the difference between the perturbed images and the original images reveals that the perturbations are more for the model after alignment phase compared to the model after refinement phase. This is anticipated as the refinement phase enforces restriction on pixels, governed by the most discriminative teacher saliency pixels. Fig 6 provides visualizations supporting the above argument in Section 6. Here we provide more such visualizations in Fig 9.

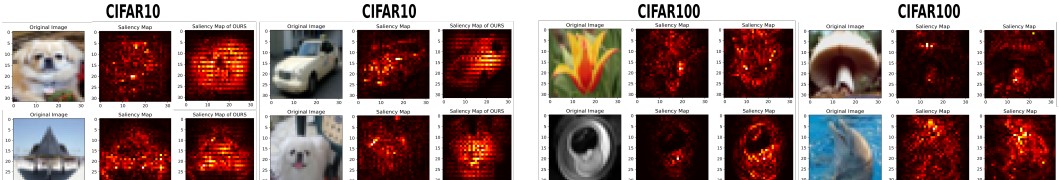

Figure 7: Visualizing effect of different teacher networks while training our student model. For each triplet of images above: *(Left)* Original image; *(Middle)* Saliency of our model trained with teacher 1 (adversarially trained on another dataset, finetuned on clean data of given dataset); *(Right)* Saliency of our model trained with teacher 2 (adversarially trained on given dataset)

Figure 8: Visualizing effect of different teacher networks while training our student model with images from CIFAR10 dataset. For each triplet of images above: *(Left)* Original image; *(Middle)* Saliency of our model trained with teacher 1 (adversarially trained on another dataset, finetuned on clean data of given dataset); *(Right)* Saliency of our model trained with teacher 2 (adversarially trained on given dataset)

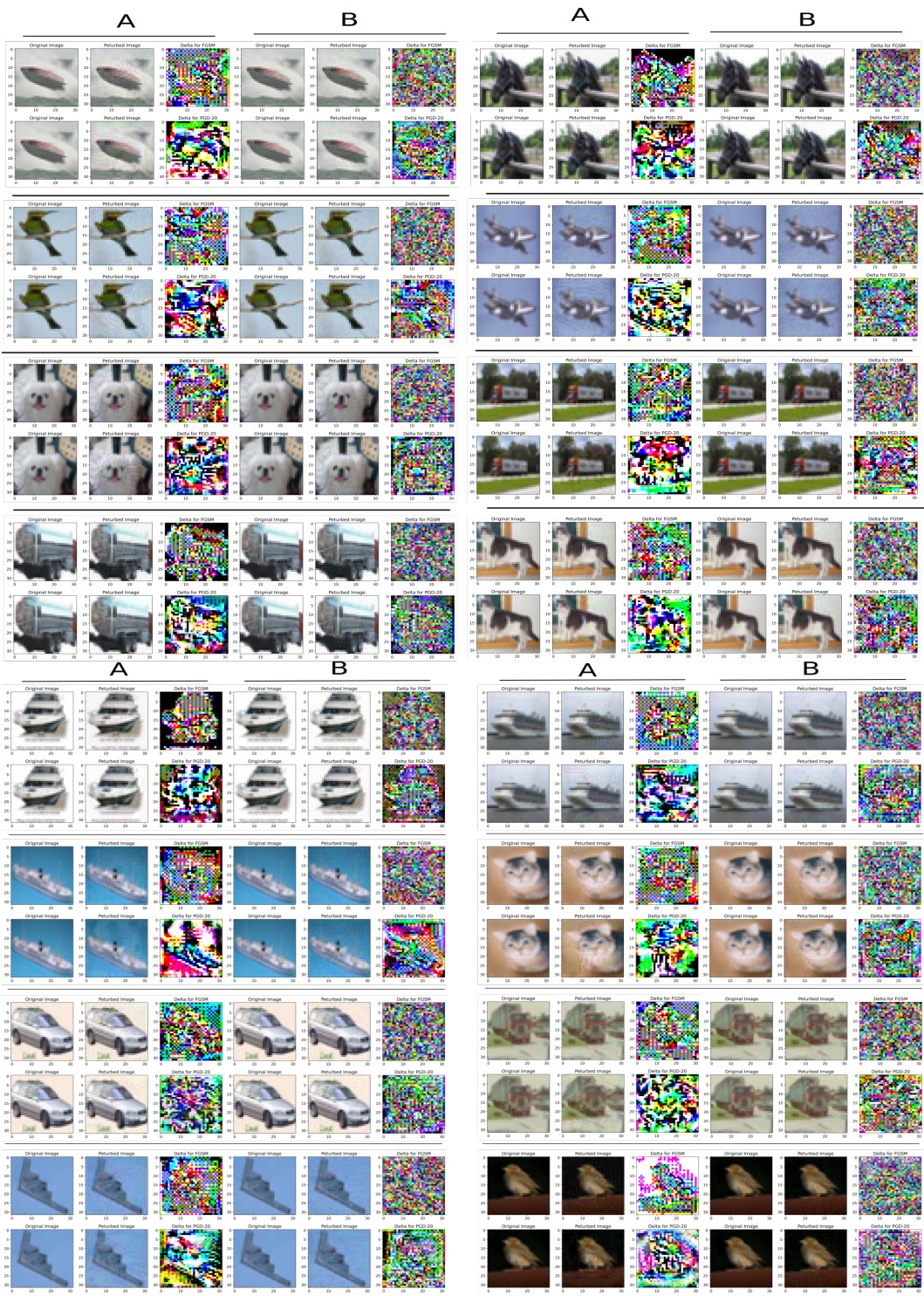

Figure 9: Effect of curriculum learning through visualizing perturbed images when the models are attacked by FGSM and PGD-20. For each group of 12 images, the top-left and bottom-left triplet corresponds to the actual image, perturbed image and net change (delta) in pixels when a model obtained after 1st training phase is attacked using FGSM and PGD-20 method respectively. Similarly, the top-right and bottom-right triplets corresponds to the same set of images obtained using a model obtained after 2nd phase of training. Here A and B represent images with models obtained after first and second phase respectively.