# OpenReview forum: "Adversarial Robustness without Adversarial Training: A Teacher-Guided Curriculum Learning Approach"
_NeurIPS.cc/2021/Conference — NeurIPS 2021 Poster_

### Official Review · Reviewer_Wty9 · 2021-07-07

**Rating:** 7
**Confidence:** 4

**Summary:**

The paper proposes a non-iterative adversarial defense. The method trains networks to have similar saliency maps as a teacher model. The authors show empirically that their method outperforms baseline defenses while requiring less training time.

**Limitations And Societal Impact:**

The authors adequately describe the impact and limitations of the work in the broader impacts section.

**Main Review:**

**Originality**
The method proposed in this paper is similar to other non-iterative adversarial training methods as mentioned by the authors in section 3. The exact methodology differs from prior work, which allows the authors to achieve better results that outperform baselines.

**Quality**
The method is not motivated theoretically, so its effectiveness must be demonstrated through experiments. The experiments are particularly important, since the paper claims to present a state-of-the-art defense. There are some potential issues with the experiments, although they may be resolvable:
1) TRADES has a lower accuracy under AutoAttack in Table 1 compared to what is reported in the AutoAttack paper. Is there a reason for this?
2) How does the proposed method compare to the state-of-the-art defenses on CIFAR-10 (particularly defenses incorporating unlabeled data) reported in the AutoAttack paper (in Table 2 of that paper)? In particular, is the proposed method the state-of-the-art defense on CIFAR-10 for 8/255 $\ell_\infty$ perturbation sizes?
3) The training time and memory consumption of the proposed method vs. baselines do not appear to be quantified anywhere. This is very important to include since computational efficiency is one of the main advantages of this approach.
4) CIFAR and TinyImagenet are good choices for evaluating adversarial defenses. However, training a robust model on ImageNet would be ideal.
5) Are both the discriminator-based and L2 saliency losses necessary to achieve robustness? In other words, how does the method perform when $\beta=0$ or $\gamma=0$?

Resolving these issues would *significantly* improve the quality of this paper.


**Clarity**
The paper is generally well organized and the writing is clear. Concepts are introduced with the appropriate context and experiments are clearly described.


**Significance**
If the issues with experimental results are resolved, this paper could represent a major contribution to the research community since it can achieve state-of-the-art performance with an order of magnitude lower computational cost than current methods.

**Time Spent Reviewing:**

2

---

> ### Author Response · Authors · 2021-08-11
> **Response to Reviewer Wty9**
>
> Thank you for the valuable comments and suggestions. We are encouraged to see your appreciation on the writing style, elucidation of the experiments, and contextual significance of concepts. We are grateful to you for acknowledging this work as a major contribution to the research community. We answer all these concerns below.
>
> > **Q1**: TRADES has a lower accuracy under AutoAttack in Table 1 compared to what is reported in the AutoAttack paper. Is there a reason for this?
>
> **A1**: We reported the accuracy numbers from the AutoAttack paper. Please refer to the “AA” column in Table 2 of the paper which shows that the robust accuracy of TRADES under AutoAttack is 53.08%.
>
> > **Q2**: How does the proposed method compare to the state-of-the-art defenses on CIFAR-10 reported in the AutoAttack paper? In particular, is the proposed method the state-of-the-art defense on CIFAR-10 for 8/255 $l_\infty$ perturbation sizes?
>
> **A2**: Our comparisons with other methods on AutoAttack (AA) is shown in Table 1 (please see rightmost column). Yes, our results are with the $L_{\infty}$ attack and $\epsilon$=8/255. We achieved state-of-the-art results on natural and adversarial accuracies against FGSM and PGD attacks, and near to state-of-the-art performance in case of AutoAttack, as shown in the table. (We note that our method does not use any additional/unlabeled data, and we hence don't compare with other methods that use unlabeled data, to maintain fair comparison).
>
> > **Q3**: The training time and memory consumption of the proposed method vs. baselines do not appear to be quantified anywhere.
>
> **A3**: **Time Consumption**: We mentioned the number of iterations required for our method in L244 of our paper. As our method is non-iterative and each step of our training process is different from the iterative AT framework, we had not included this. We agree it helps convey our point better though. The total compute time for completion of training for our model is **~6 hours** (including phase 1 and 2), while the Adversarially Trained (AT) model takes **~30 hours** for CIFAR10 with WRN32-10 architecture (our method takes approx 20% of the training time of an AT model).
> **Memory Consumption**: For both CIFAR10 and CIFAR100, our model training takes **~11GB** of GPU with batch size of 32, whereas AT training takes **~7GB** of GPU with the same batch size. For TinyImageNet, our model and AT trained model take almost similar amounts of memory i.e. **~11GB** and **~7GB** respectively with batch size of 16. We note that our model's memory requirements fit into one commonly available single-precision GPU's RAM (e.g. GTX 1080Ti).
>
> > **Q4**: CIFAR and TinyImagenet are good choices for evaluating adversarial defenses. However, training a robust model on ImageNet would be ideal.
>
> **A4**: Thank you for the suggestion. Due to the short turnaround of this response and the computational requirements of results on ImageNet, we could not obtain the results. We will be very happy to add these results in the final version. We note however that CIFAR10 and CIFAR100 are most commonly used by related relevant work (such as [3], [4], [5], [6], [7]). Fewer methods (such as [1], [2]) even benchmark on TinyImagenet, and that's why we had restricted to TinyImagenet dataset for comparison purposes.
>
> > **Q5**: Are both the discriminator-based and L2 saliency losses necessary to achieve robustness? In other words, how does the method perform when $\beta$=0 or $\gamma$=0?
>
> **A5**: $L_{diff}$ loss is used to generate better saliency maps that match teacher saliency as given in L161-162 of our paper. We added $L_{robust}$ to minimize the $l_2$ distance between teacher and student saliency as in L165-166 of our paper. Hence, both these losses together help serve our overall objective, and they achieve global minimum individually when both saliency maps match. Adding both the losses helped to get stronger signals and attain better performance compared to using any one of them, which is also supported by our experimental findings below.
>
> We showed the relative effect of $\gamma$ (without changing the value of $\beta$) for CIFAR-10 and CIFAR-100 in Fig 5 of the main paper. Specifically, we show performance of our method with $\beta$=0 or $\gamma$=0 in the below table for CIFAR-10:
>
> |   | Natural  | PGD-20 |
> |---|:---:|:---:|
> | $\beta$=2 & $\gamma$=20| **90.63** | **59.59** |
> | $\beta$=0 & $\gamma$=20  | 87.08  | 54.83 |
> | $\beta$=2 & $\gamma$=0  | 84.79 | 41.43 |
>
> References:
> [1]: Chan, Alvin and Tay, Yi and Ong, Yew-Soon. "What it thinks is important is important: Robustness transfers through input gradients." CVPR2020.
>
> [2]: Cui, Jiequan and Liu, Shu and Wang, Liwei and Jia, Jiaya. "Learnable boundary guided adversarial training." https://arxiv.org/abs/2011.11164.
>
> [3]: Wang, Yisen and Zou, Difan and Yi, Jinfeng and Bailey, James and Ma, Xingjun and Gu, Quanquan. "Improving adversarial robustness requires revisiting misclassified examples."ICLR2019
>
> [4]: Wu, Dongxian and Xia, Shu-Tao and Wang, Yisen. "Adversarial weight perturbation helps robust generalization". NeurIPS2020
>
> [5]: Zhang, Hongyang and Yu, Yaodong and Jiao, Jiantao and Xing, Eric and El Ghaoui, Laurent and Jordan, Michael. "Theoretically principled trade-off between robustness and accuracy". ICML2019
>
> [6]: Madry, Aleksander and Makelov, Aleksandar and Schmidt, Ludwig and Tsipras, Dimitris and Vladu, Adrian. "Towards deep learning models resistant to adversarial attacks". ICLR2018
>
> [7]: Zhang, Jingfeng and Zhu, Jianing and Niu, Gang and Han, Bo and Sugiyama, Masashi and Kankanhalli, Mohan. "Geometry-aware instance-reweighted adversarial training." ICML2021

---

> > ### Comment · Reviewer_Wty9 · 2021-08-28
> > **Thank you for your detailed response!**
> >
> > I find the authors' responses regarding their experiments valuable and convincing, and I have significantly raised my rating accordingly.

---

> > > ### Author Response · Authors · 2021-08-30
> > > **Thank you**
> > >
> > > Dear Reviewer Wty9,
> > >
> > > Thank you for your response. We are motivated to see that you found our paper deserves to be accepted, and that the paper could represent a major contribution to the research community. Most concerns of reviewers revolved around certain clarifications in the methodology and experiments, each of which we have carefully addressed in our responses and will integrate in our final paper.

---

> ### Comment · Area_Chair_kEfJ · 2021-08-11
> **discussion**
>
> Does the clarifications brought by the authors change you mind. In particular you write that answering 1-4 would significantly improve your opinion and they are providing answers for 1 and 3, but what do you think of the answer to  2 and do you consider the promise for 4 enough?

---

### Official Review · Reviewer_b3WR · 2021-07-15

**Rating:** 5
**Confidence:** 4

**Summary:**

The paper proposes an AT based defense method, which only focuses on the salient image regions to perturb and attack. Saliency is computed from a pertained teacher model; and is gradually reduced during the training (hence the name curriculum learning). A two phase approach is proposed: phase-1 computes the saliency map; and phase-2 generates adversaries by altering the salient pixels only (by gradually reducing the salient pixels from 90% upto 50%). Experiments are given on CIFAR-10/100 and TinyImageNet.


**Limitations And Societal Impact:**

Conclusion section of the paper briefly mentions pros and cons of the research.

**Main Review:**

Can authors validate their assumption that perturbing the salient/foreground image pixels results in quick strong adversaries, by considering a pertained segmentation network, and restricting the perturbations to salient regions, and providing a comparison with perturbing all pixels (vs salient pixels)?

An underlying assumption is that the model trained using the loss in phase 1 (Eq 13) generates saliency map which can localise the object. I was expecting some visualisations of that. Fig4&Fig7 have a few samples, but they’re too few. Can authors show some more examples in the supplementary? Also, can they quantify (e.g., Jaccard Index) on how good are the generated saliency maps; for a dataset with ground truth segmentation masks?

A major claim of the paper is its compute efficiency. I did not see any comparison with existing AT based defense methods for comparison in terms of compute time; number of iterations required. The proposed method has an additional step (phase-1 of the method), which isn’t the case for most sota AT based defences. A fair comparison is required to substantiate the efficiency claims made in the paper.


**Time Spent Reviewing:**

4

---

> ### Author Response · Authors · 2021-08-11
> **Response to Reviewer b3WR**
>
> We thank the reviewer for the insightful comments and suggestions. We reply to all your comments below.
>
> > **Q1**: Can authors validate their assumption that perturbing the salient/foreground image pixels results in quick strong adversaries?
>
> **A1**: There may be a mild misunderstanding here which we clarify. We do not claim that attacks on salient/foreground pixels are stronger; we rather claim that forcing the attacker to only focus on the salient/foreground object pixels in an image makes it harder to attack a model. To expand on this, we aim to achieve robustness by restricting the attack space to the most discriminative pixels of the object i.e. by considering only the salient features of the object for perturbation. This strategy ensures that if the attacker tries to perturb the image, s/he might either obfuscate the object, in which case a change of class or low probability is desirable or be unsuccessful due to the reduced attack space. We enforce this condition through our model training. This implies that trying to fool the model trained by our method, would come from perturbations mostly in the salient object pixels. We show this effect in Figs 6 and 8 of our paper, where the perturbations after the second phase are more restricted to the object pixels compared to the perturbations after the first phase.
>
> > **Q2**: An underlying assumption is that the model trained using the loss in phase 1 (Eq 13) generates saliency map which can localise the object. I was expecting some visualisations of that. Fig4&Fig7 have a few samples, but they’re too few. Can authors show some more examples in the supplementary? Also, can they quantify (e.g., Jaccard Index) on how good are the generated saliency maps?
>
> **A2**: As our saliency alignment is enforced throughout the training process, our model generates better saliency maps after the second phase compared to the first phase. We have provided saliency maps generated by our model after phase 1 on this [link](https://anonymous.4open.science/r/Rebuttal-link-1-F748/OUR_phase1_localize.png) (anonymized link shared only as a response to this question). We have reported samples of saliency maps after phase 2 in Figs 4 and 7, and more such examples are given on this [link](https://anonymous.4open.science/r/Rebuttal-link-2-8774) (anonymized link shared only as a response to this question). The maps are sequenced exactly as in Figs 4 and 7 of the main paper.
>
> As the ground truth segmentation masks for CIFAR10 and CIFAR100 aren’t available, we conducted a human Turing test to check the goodness of the saliency maps by asking 10 human users to pick a single winning saliency map that was most true to the object in a given image among saliency maps generated by AT [Madry et. al.] and our model attained after phase-1 training. The winning rates (in same order: AT, Ours) were: CIFAR10 **[20%,80%]** and CIFAR100 **[30%,70%]**, supporting our claim. For TinyImageNet, we calculated the Jaccard Index between ground truth segmentation masks and the saliency maps generated by AT [Madry et. al.] as well as our model after phase-1 on its test dataset. We obtained Jaccard Index values as **0.63** and **0.74** for AT and our method, respectively. All these results demonstrate that the saliency maps, generated by our model after first phase, is better than an AT model in terms of localizing the objects in the image.
>
> > **Q3**: A major claim of the paper is its compute efficiency. I did not see any comparison with existing AT based defense methods for comparison in terms of compute time; number of iterations required.
>
> **A3**: We mentioned the number of iterations required for our method in L244 of our paper. As our method is non-iterative and each step of our training process is different from the iterative AT framework, we had not included this. We agree it helps convey our point better though. The total compute time for completion of training for our model is **~6 hours** (including phase 1 and 2), while the Adversarially Trained (AT) model takes **~30 hours** for CIFAR10 with WRN32-10 architecture (our method takes approx 20% of the training time of an AT model).

---

> > ### Author Response · Authors · 2021-08-30
> > **Follow-up to our response**
> >
> > Dear Reviewer b3WR,
> >
> > We have responded to each of your individual concerns to the best of our knowledge. We request you to please look at our response, and let us know if any further information/clarification is required that can further improve the usefulness and presentation of our work.

---

### Official Review · Reviewer_hwPL · 2021-07-16

**Rating:** 6
**Confidence:** 3

**Summary:**

This paper proposes a non-iterative method to improve adversarial robustness. Compared to mainstream AT baselines, the proposed method avoids iterative gradient calculations, so as to be time-saving. By involving attribution maps and perturbing the allowed set of pixels, the proposed method can achieve better performance. After sufficient experiments, the proposed method beats all baselines.

**Limitations And Societal Impact:**

The authors adequately addressed the limitations and potential negative social impact of their work.

**Main Review:**

Pros:

Compared to traditional multi-step AT, this paper has a novel solution Teacher-guided Saliency-based Robust Training and clear explanation.

The paper conducted extensive experiments to validate the effectiveness of the proposed method, including different datasets, attack methods, and baselines.

This paper has startling performance, especially on natural accuracy in the case of improvement on adversarial accuracy.

Cons:
My most concern is about the organized and written style.
-The title ”Adversarial Robustness without Adversarial Training: A Teacher-Guided Curriculum Learning Approach” does not fit the insights.
The method that “Teacher-guided Saliency-based Robust Training” is also a solution of training. By the way, although Adversarial Training is the mainstream to achieve robustness, there are still other ways such as Obfuscated Gradients, “Adversarial Robustness without Adversarial Training” need rethinking.

-The main effect mentioned in the abstract is time-saving, however, there is a Teacher model, and regardless of the training time of the teacher model is unfair. Suggest this paper can reorganize its contribution.

-For adversarial-robust models trained by baselines, it is feasible to use any attack method to test its performance, while too much NA is in the experimental tables, bad for comparison.



**Time Spent Reviewing:**

5

---

> ### Comment · Area_Chair_kEfJ · 2021-08-11
> **discussion**
>
> Autors didn't answered the questions of reviewers, but you had some positive opinion on the paper. Would you like to champion the paper?
>
> Edit: authors now have poster their answers, I think this was because the new system plans to have a rolling discussion with authors which is a new thing. Please don't consider they were late on rebuttal. Yet the question of supporting the paper remains valid! Does their answers and other reviews shift your opinion?

---

> ### Author Response · Authors · 2021-08-11
> **Response to Reviewer hwPL**
>
> Thank you for the sincere review and suggestions. We are inspired that you found our work novel with clear explanations and extensive experimentations helpful to show our method’s efficacy. We reply to all the comments below.
>
> > **Q1**: The title ”Adversarial Robustness without Adversarial Training: A Teacher-Guided Curriculum Learning Approach” does not fit the insights.
>
> **A1**: We appreciate the suggestions made regarding the title of this work. We agree that deviating from mainstream robust training approach shouldn’t be reflected in the title as other training methods are available. We wanted to emphasize on the curriculum style learning and agree with the title “Teacher-guided Saliency-based Curriculum Style Robust Training”. We will make this change.
>
> > **Q2**: The main effect mentioned in the abstract is time-saving, however, there is a Teacher model, and regardless of the training time of the teacher model is unfair.
>
> **A2**: We used an off-the-shelf adversarially trained teacher (with no additional training time at our end) for our model training and accordingly claimed the training time of our model, which is ~20% of the time required for AT model training. We agree however that this could be viewed as a way to improve existing AT models with a little extra effort (upto 20%) to obtain significant gains over existing methods, and we will revise our claims to make this clear. We reiterate that the gains obtained using our method are significant and non-trivial. Also, our method works with other teacher models too (as shown in Tables 3 and 4) but the best results are achieved with an adversarially trained teacher.
>
> To study further, we compared our method with recent work which aim to improve robustness by adversarial pretraining and fine-tuning on the same dataset. Comparative results on CIFAR-10 are given below, validating the usefulness of our method:
>
> | Model  | Nat.  | PGD-20 |
> |---|:---:|:---:|
> |**OUR**| **90.63** | **59.59** |
> | Hendrycks et. al. [1] | 87.10  | 57.40 |
> | Chen et. al. [2] | 86.04  | 54.64 |
>
> > **Q3**: For adversarial-robust models trained by baselines, it is feasible to use any attack method to test its performance, while too much NA is in the experimental tables, bad for comparison.
>
> **A3**: Apart from FGSM and PGD-20, we also wanted to compare against the few methods which provided their performance for PGD-5 and PGD-10. We’ll remove the columns for PGD-5 and PGD-10 from the main table and add another table comparing only on these settings without NA values.
>
> References:
> [1]: Hendrycks, Dan and Lee, Kimin and Mazeika, Mantas. “Using pre-training can improve model robustness and uncertainty.” ICML2019.
>
> [2]: Chen, Tianlong and Liu, Sijia and Chang, Shiyu and Cheng, Yu and Amini, Lisa and Wang, Zhangyang. “Adversarial robustness: From self-supervised pre-training to fine-tuning.” CVPR2020.

---

> > ### Author Response · Authors · 2021-08-30
> > **Follow-up to our response**
> >
> > Dear Reviewer hwPL,
> >
> > We have responded to each of your individual concerns to the best of our knowledge. We request you to please look at our response, and let us know if any further information/clarification is required that can further improve the usefulness and presentation of our work.

---

### Official Review · Reviewer_EoWP · 2021-07-17

**Rating:** 5
**Confidence:** 4

**Summary:**

The paper proposes a two phase non-iterative approach for efficient training of robust models.

The first phase is a distillation phase which trains a neural network while enforcing the gradients of the loss w.r.t. to the input image of the student network to be indistinguishable to that same gradient for the teacher network with the goal of enforcing the saliency maps of the teacher and student networks to be “similar”. In addition to the binary cross-entropy GAN-type loss, the paper also explicitly promotes the two saliency maps to be similar by adding an $\ell_2$ regularization on their differences.

The second phase, which is called model refinement, is a curriculum learning based adversarial training fine-tuning step. At every step of this phase, the attacker is limited to only perturbing the top X% of pixels from the teachers saliency. As training progresses, the attacker is more limited (i.e., X gets smaller).

**Main Review:**

The problem considered is an important problem and a well motivated problem: How can we efficiently train a robust network which maintains high natural accuracy? The authors have done a good job in phrasing and motivating the problem. The empirical results reported look good. However, I do have some intuition/theoretical questions.

One of the main premises of the paper is that it achieves robustness very fast (efficiently) since it is a non-iterative method/compared to iterate methods. However, given that during the first phase, the best performing models of this methods utilize iteratively trained robust networks trained on the same dataset (from Tables 3, and 4), it seems like that this method should either contain a phase 0: adversarially train a robust network, or, the claims should be a bit toned down to efficiently improving robustness of robust networks. And for the later case, the method would have a better / fairer comparison if compared to other methods which are based on either pertaining and transfer learning.

The first phase of this approach seems pretty complicated and while it has motivation, it leaves me with many questions:
1- Why do we need both L_diff and L_robust for the alignment loss? From the paper (line 163), the global minimum of L_robust is achieved when the saliency maps match, which implies that L_diff is indeed what is needed. I do wonder that, from a theoretical point of view, what does L_robust do?

2- point 1 even pushed further, why do we need phase 1 completely? Doesn’t using the teacher model (specially for cases where the teacher is trained on the same dataset as the student), result in minimum loss for the first phase? If we drop the first phase, then the second phase is similar to a LwF penalty added to robust training.

3- what is the value of loss for each of the terms after phase1 as completed? What would the value of loss be if we simply plug the Madry WRN32-10 robust net for CIFAR-10?

For the second phase, the curriculum learning is designed to make the attacker weaker, which intuitively *may* result in lower robustness (but at the same time may be beneficial for natural accuracy), do the authors have an intuition why gradually making the attacker weaker during training, can result in an improvement in robustness?
Also, what is the intuition behind still using the teachers saliency? The saliency consistency regularizations for phase 2 resembles the learning without forgetting loss --  It seems like that you refine the net but still maintain the saliency map of the robust net from phase 1 (or phase 0)?

From results in Table1, it seems that for the curriculum based methods, the strongest attacks are ether AA & CW. I wonder, how the CIFAR-100 models (Table2) would do with these attacks? Also, comparing PGD-5, to PGD-10, and then PGD-10 to PGD-20, it seems like the methods robustness may not have yet plateaued under the PGD attack, I do wonder what is the robustness if we increase the number of iterations further?

Other:
Small typo: In line 124, tern -> term.
Figure 6 contradicts some recent papers which claim that the saliency maps of robust nets are more aligned with human perception. When going to phase 2 (B) from phase (A), if we are improving robustness, it seems like we are making the saliency map more noisy, any intuition on this?


**Time Spent Reviewing:**

4

---

> ### Author Response · Authors · 2021-08-11
> **Response to Reviewer EoWP**
>
> Thank you for your thorough and insightful comments. We are encouraged to see that you appreciated our work as well-motivated on the important problem of training robust networks without sacrificing natural accuracy. We reply to all the points below.
>
> > **Q1**:  It seems like that this method should either contain a phase 0: adversarially train a robust network, or, the claims should be a bit toned down to efficiently improving robustness of robust networks. And for the later case, the method would have a better / fairer comparison if compared to other methods which are based on either pertaining and transfer learning.
>
> **A1**: We achieved significant performance improvement by using a teacher network which is adversarially trained (AT) on the same dataset, as shown in Tables 3 and 4 of our paper. For this purpose, we used an off-the-shelf adversarially trained model (with no additional training time at our end). We agree however that this could be viewed as a Phase0 as suggested, and will revise our claims accordingly. Our method though is not necessarily a derivative of AT, rather the motivation of our first phase training is enforcing feature similarity with a suitable teacher network to train the student model from scratch (we show results with other teachers in Tables 3 and 4). In the case that the teacher is adversarially trained, one could view our work as a way to improve existing AT models with a little extra effort (upto 20%) to obtain significant gains. We reiterate that the gains obtained using our method however are significant and non-trivial.
>
> To study further, we compared our method with recent work which aim to improve robustness by adversarial pretraining and fine-tuning on the same dataset. Comparative results on CIFAR-10 are given below, validating the usefulness of our method:
>
> | Model  | Nat.  | PGD-20 |
> |---|:---:|:---:|
> |**OUR**| **90.63** | **59.59** |
> | Hendrycks et. al. [1] | 87.10  | 57.40 |
> | Chen et. al. [2] | 86.04  | 54.64 |
>
> > **Q2**: Why do we need both $L_{diff}$ and $L_{robust}$ for the alignment loss?  I do wonder that, from a theoretical point of view, what does L_robust do?
>
> **A2**: $L_{diff}$ loss is used to generate better saliency maps that match teacher saliency as given in L161-162 of our paper. We added $L_{robust}$ to minimize the $l_2$ distance between teacher and student saliency as in L165-166 of our paper. Hence, both these losses together help serve our overall objective, and they achieve global minimum individually when both saliency maps match. Theoretical justification of attaining global minimum for $L_{robust}$ loss is shown in *Theorem 3.1* of [3]. Adding both the losses helped to get stronger signals and attain better performance compared to using any one of them, which is also supported by our experimental findings (see [response to Reviewer Wty9](https://openreview.net/forum?id=MqCzSKCQ1QB&noteId=nRMkBi8aE7)).
>
> > **Q3**: If we drop the first phase, then the second phase is similar to a LwF penalty added to robust training.
>
> **A3**: Our second phase is a continuation of the first phase with added curriculum learning based on limiting the object pixels to be perturbed, that change the class score. The key idea, especially the curriculum style learning, is different from adding just a *LwF* penalty to robust training. One could view the second phase of our training though as “*Curriculum Learning + LwF loss*". Here, alignment loss tries to make sure that the model should not forget the knowledge of the full object, which was already learned in the first phase. From this perspective, we can say alignment loss at the second phase plays a similar role as *LwF*. But the way we enforce the *LwF* is completely different from earlier efforts such as [4]. This is explained in L308-315 in the main paper.
>
> > **Q4**: Why do we need phase 1 completely? What is the value of loss for each of the terms after phase1 as completed? What would the value of loss be if we simply plug the Madry WRN32-10 robust net for CIFAR-10?
>
> **A4**: This is an interesting question. Ideally, after the first phase, if we simply plug the Madry WRN32-10 robust net for CIFAR-10, then the model would incur zero alignment loss. But CE loss of such a model would be more compared to the model trained using our method. After the first phase, We achieve 1% to 2% improvement in natural as well as adversarial accuracy using our model compared to Madry’s robust model. This is shown in Tables 8 and 9 of our supplementary section. For comparing with AT, we also computed each loss term after phase 1:
>
> |   | CE loss  | Alignment loss  |
> |---|:---:|:---:|
> |Our Model| **0.3594**  |  0.009  |
> | AT Model  | 0.4956  | **0.0**  |
>
> Moreover, in order to show the importance of the first phase training, we train the second phase, starting from Madry’s Robust model, instead of the model achieved after the first phase. Our experimental findings suggest that robustness of the final model (achieved after second phase of training ) is much less if we use Madry’s model (AT) directly. Experimental results on CIFAR-10 are below:
>
> | Model  | Nat.  | FGSM  | PGD-20 |
> |---|:---:|:---:|:---:|
> |Phase-1 : **OUR** --> Phase-2 : **OUR**| **90.63** | **67.84** | **59.59** |
> | Phase-1:  **AT**    -->  Phase-2 : **OUR**  | 84.47  | 57.19  | 51.10  |
>
> > **Q5**: Do the authors have an intuition why gradually making the attacker weaker during training, can result in an improvement in robustness?
>
> **A5**: The intuition behind the refinement phase follows the intuition of curriculum learning -- that gradually providing information to a model helps it learn better. Our objective is that the set of pixels, to be perturbed for changing class prediction, should belong to the most discriminative parts of the image. The curriculum style learning enforces this gradually starting from full image to comparatively more discriminative parts of the objects at every stage. This improves robustness as the model learns to restrict the attack space to the discriminative parts of the image when the image is perturbed to change the class decision. This is analogous to making the attack weaker since the perturbation will be concentrated more in the discriminative parts and less in the other parts. The importance of curriculum style learning is explained in Section C of our supplementary section (L495-504). Our experimental findings also justify the efficacy of curriculum learning, as shown in Tables 10 and 11 in the supplementary section.
>
> > **Q6**: What is the intuition behind still using the teachers saliency? The saliency consistency regularizations for phase 2 resembles the learning without forgetting loss -- It seems like that you refine the net but still maintain the saliency map of the robust net from phase 1 (or phase 0).
>
> **A6**: Learning to limit the pixels to be perturbed, to the discriminative parts of the object in the second phase while maintaining knowledge about features of the whole object is the motivation behind keeping the consistency regularizer during the second phase. The saliency consistency regularization helps to maintain the saliency from the teacher network throughout the training process. We explained this intuition in L216-223 of the main paper. Hence our motivation is similar to using the learning without forgetting loss, but the purpose and way of implementing it is indeed different.
>
> > **Q7**: How the CIFAR-100 models (Table2) would do with AA & CW attacks? It seems like the methods robustness may not have yet plateaued under the PGD attack, what is the robustness if we increase the number of iterations further?
>
> **A7**: We achieved **27.92%** & **29.76%** with AA & CW attacks for CIFAR-100 models, which we’ll add in the final version. We achieve below results with increasing number of iterations of PGD attack on CIFAR-10, which shows that our method is plateaued under the PGD attack. We added results with PGD-20 for comparison purpose.
>
> | Model  | PGD-20  | PGD-50 | PGD-70  | PGD-100 |
> |---|:---:|:---:|:---:|:---:|
> |**OUR**| **59.59** | **58.50** | **58.22** | **58.04** |
>
>
> > **Q8**: Small typo: In line 124, tern -> term
>
> **A8**: We’ll correct it in the modified draft.
>
> > **Q9**: Figure 6 contradicts some recent papers which claim that the saliency maps of robust nets are more aligned with human perception. When going to phase 2 (B) from phase (A), if we are improving robustness, it seems like we are making the saliency map more noisy, any intuition on this?
>
> **A9**: Figure 6 visualizes perturbed images when the models are attacked by FGSM and PGD-20; the maps show the net change ($\delta$) in pixels, **not saliency maps**. For each group of 12 images, the top-left and bottom-left triplets correspond to the actual image, perturbed image and net change ($\delta$) in pixels, when a model obtained after 1st training phase is attacked using FGSM and PGD-20 method respectively. Similarly, the top-right and bottom-right triplets correspond to the same set of images obtained using a model obtained after second phase of training. Here A and B represent images with models obtained after first and second phase respectively. This shows effectiveness of the second phase training in restricting the change in pixels ($\delta$) due to attacks. We'd be happy to clarify this further if required.
>
> References:
> [1]: Hendrycks, Dan and Lee, Kimin and Mazeika, Mantas. "Using pre-training can improve model robustness and uncertainty." ICML2019.
>
> [2]: Chen, Tianlong and Liu, Sijia and Chang, Shiyu and Cheng, Yu and Amini, Lisa and Wang, Zhangyang. "Adversarial robustness: From self-supervised pre-training to fine-tuning." CVPR2020.
>
> [3]: Chan, Alvin and Tay, Yi and Ong, Yew Soon and Fu, Jie. “Jacobian adversarially regularized networks for robustness.” ICLR2020.
>
> [4]: Shafahi, Ali and Saadatpanah, Parsa and Zhu, Chen and Ghiasi, Amin and Studer, Christoph and Jacobs, David and Goldstein, Tom. “Adversarially robust transfer learning.” ICLR2020.

---

> > ### Author Response · Authors · 2021-08-30
> > **Follow-up to our response**
> >
> > Dear Reviewer EoWP,
> >
> > We have responded to each of your individual concerns to the best of our knowledge. We request you to please look at our response, and let us know if any further information/clarification is required that can further improve the usefulness and presentation of our work.

---

> > > ### Comment · Reviewer_EoWP · 2021-08-31
> > > **Thanks for the rebuttal**
> > >
> > > Thank you for the effort taken to write the rebuttal. I reviewed your responses in detail and they addressed some of my concerns-- I greatly appreciate the time put to write the response. However, I still don't find the find the choices very intuitive and hence am inclined to maintain my score. Please find some additional comments below:
> > >
> > >
> > >  --------
> > > A1-1: Our method though is not necessarily a derivative of AT, rather the motivation of our first phase training is enforcing feature similarity with a suitable teacher network to train the student model from scratch (we show results with other teachers in Tables 3 and 4). In the case that the teacher is adversarially trained, one could view our work as a way to improve existing AT models with a little extra effort (upto 20%) to obtain significant gains.
> > > R1-1: The part which was raised is that how important it is for the "suitable teacher" to be adversarially trained. Results in Tables 3 and 4 still all use adversarially trained teachers. If the claim is to achieve very fast robustness, then It would be interesting to see what the robustness is under different choices of less expensive phase0 models.
> > > A1-2: To study further, we compared our method with recent work which aim to improve robustness by adversarial pretraining and fine-tuning on the same dataset.
> > > R1-2: Minor: The work of Hendrycks et al., pretraines on another dataset. Other: In their appendix, they illustrate the true robustness by measuring against multiple random restarts which is probably a better measure for comparison.
> > >
> > > -------------
> > > A2: Adding both the losses helped to get stronger signals and attain better performance compared to using any one of them, which is also supported by our experimental findings (see response to Reviewer Wty9).
> > > R2: In the paper, a lot of effort has been put in showing the variance and sensitivity of the model to different choices of $\gamma$ . Thus, it is not very convincing from an experimental point of view that both losses are indeed needed by illustrating the performance of having one loss with one choice of parameter (it might actually be the case but I can't confidently conclude that yet.)
> > >
> > > -------
> > > R4: Thanks for these set of experiments. It is interesting how the second phase of experiments results in such a boost for a model trained with phase1 as opposed to another good candidate for phase 1 (based on the objective/loss). I'd like to re-iterate that the numbers look good for the optimal set of hyper-parameters, but from a intuition point of view, I don't fully understand why replacing the phase1 model with the teacher adversarially trained model on the same dataset would result in such a big difference and why the results are so sensitive.

---

> > > > ### Author Response · Authors · 2021-09-01
> > > > **Follow-up response to Reviewer EoWP**
> > > >
> > > > Thank you for sharing your thoughts.
> > > >
> > > > > R1-1: ...how important it is for the "suitable teacher" to be adversarially trained. Results in Tables 3 and 4 still all use adversarially trained teachers. If the claim is to achieve very fast robustness, then It would be interesting to see what the robustness is under different choices of less expensive phase0 models.
> > > >
> > > > Tables 3 and 4, in fact, show the effect of using less expensive teacher models, comprising of adversarial training on smaller datasets followed by naturally fine-tuning on the actual dataset (please see rows corresponding to teachers "A" and "C" in Tables 3 and 4 respectively). If we go more extreme and train using only a naturally trained model, the results are expectedly poor (which we observed in our studies too). As the saliency map of a naturally trained teacher can be noisy, applying our curriculum learning strategy with a naturally trained teacher does not result in capturing "good" features and hence not as well-suited for our purpose. (Adversarially trained and naturally trained are the natural choices for teacher models in our setting -- if there are other specific teacher models we should attempt, please let us know, and we'd be happy to include them.) Following our claim that the goodness of the saliency map of the teacher model is important for our model performance, we achieve best results with a teacher model, which is adversarially trained on the actual dataset.
> > > >
> > > >
> > > > > R1-2: Minor: The work of Hendrycks et al., pretrains on another dataset. Other: In their appendix, they illustrate the true robustness by measuring against multiple random restarts which is probably a better measure for comparison.
> > > >
> > > > We had studied our model with random restarts and found that our results only vary by a negligible range (~+-0.02%) compared to the results we achieve without random restart, which we already reported in our paper. We did not report the results with random restarts for this reason, and will be happy to include it if useful. Please note that our method not only outperforms Hendrycks et al significantly, but also achieves state-of-the-art adversarial accuracy even considering baselines with random restart setup. Following mainstream papers on adversarial robustness, we reported results without random restart, but will be happy to include our results on random restart setup as well.
> > > >
> > > >
> > > > > R2: In the paper, a lot of effort has been put in showing the variance and sensitivity of the model to different choices of $\gamma$. Thus, it is not very convincing from an experimental point of view that both losses are indeed needed by illustrating the performance of having one loss with one choice of parameter (it might actually be the case but I can't confidently conclude that yet.)
> > > >
> > > > The study of variance and sensitivity of the model to $\gamma$ was included only as an ablation to show the impact of the choice of the hyperparameter, for purposes of completeness of analysis.
> > > >
> > > > Please see A5 in our [response to Reviewer Wty9](https://openreview.net/forum?id=MqCzSKCQ1QB&noteId=nRMkBi8aE7), which shows that the best result is attained with keeping both $\gamma$ and $\beta$ compared to removing any one of them. This reinforces the importance of having both losses in order to achieve our objective optimally. We reproduce these results for convenience below:
> > > >
> > > > We showed the relative effect of $\gamma$ (without changing the value of $\beta$) for CIFAR-10 and CIFAR-100 in Fig 5 of the main paper. Specifically, we show performance of our method with $\beta$=0 or $\gamma$=0 in the below table for CIFAR-10:
> > > >
> > > > |   | Natural  | PGD-20 |
> > > > |---|:---:|:---:|
> > > > | $\beta$=2 & $\gamma$=20| **90.63** | **59.59** |
> > > > | $\beta$=0 & $\gamma$=20  | 87.08  | 54.83 |
> > > > | $\beta$=2 & $\gamma$=0  | 84.79 | 41.43 |
> > > >
> > > >
> > > > > R4: Thanks for these set of experiments. It is interesting how the second phase of experiments results in such a boost for a model trained with phase1 as opposed to another good candidate for phase 1 (based on the objective/loss). I'd like to re-iterate that the numbers look good for the optimal set of hyper-parameters, but from a intuition point of view, I don't fully understand why replacing the phase1 model with the teacher adversarially trained model on the same dataset would result in such a big difference and why the results are so sensitive.
> > > >
> > > > As stated in L118-119 of our paper, adversarially robust models have better saliency maps (input gradients) compared to noisy saliency maps for naturally trained models, which is also supported in [5],[6] and [7]. Hence, a model trained via alignment with an adversarially trained teacher shows better robustness and interpretability of saliency maps. We reiterate that goodness of saliency maps of the teacher model plays a crucial role for our method's performance. This is enforced by alignment with an adversarially robust teacher model in our studies, which in turn helps the student's saliency map capture robust features, and this acts as a foundation for phase2. This observation is also supported empirically in Tables 3 and 4, as the best performance is reported with a fully adversarially trained teacher model on the same dataset.
> > > >
> > > > We hope we have adequately clarified your concerns now. We will be happy to answer more concerns in the discussion timeframe provided to us.
> > > >
> > > >
> > > > References:
> > > >
> > > > [5]: Etmann, Christian and Lunz, Sebastian and Maass, Peter and Sch{\"o}nlieb, Carola-Bibiane. “On the connection between adversarial robustness and saliency map interpretability.” ICML2019.
> > > >
> > > > [6]: Chan, Alvin and Tay, Yi and Ong, Yew-Soon. “What it thinks is important is important: Robustness transfers through input gradients.” CVPR2020.
> > > >
> > > > [7]: Tsipras, Dimitris and Santurkar, Shibani and Engstrom, Logan and Turner, Alexander and Madry, Aleksander. “Robustness may be at odds with accuracy.” ICLR2019.

---

### Author Response · Authors · 2021-08-11
**Common Response**

We thank the reviewers for the constructive and valuable feedback. We are encouraged to see that they found our work well-motivated in general, and that it targets an important problem of maintaining high natural accuracy with efficient robust training. We are also delighted that they felt this work has a clear explanation with extensive experiments to validate the effectiveness of the method. The precise comments greatly help improve clarifying the purpose of our work and its presentation. We address the comments to all the reviewers individually and will incorporate the feedback in the final version.

---

### Author Response · Authors · 2021-08-19
**Request for acknowledgement of rebuttal**

Dear Reviewers and Area Chairs,

The concerns about the paper raised by the reviewers -- both clarifications and additional experimental validation -- have been answered.  We have hopefully clarified all the questions in the rebuttal and will update the draft accordingly for improved comprehension. We request the reviewers to go through the responses; if there are any further questions or concerns, we'd be happy to answer them.

---

### Author Response · Authors · 2021-08-29
**Request for discussions**

Dear AC and Reviewers,

With the rebuttal/discussion period nearing a close, we once again request you to please let us know if you had further questions/clarifications. To the best of our knowledge, we have addressed all individual reviewer concerns -- both clarifications and additional experimental validation. We thank Reviewer Wty9 for the post-rebuttal response. We sincerely request other reviewers for further queries/discussions that can help quell concerns and revisit your rating, and more importantly, help us improve the paper overall.

---

### Decision · Program_Chairs · 2021-09-27

**Decision:**

Accept (Poster)

**Comment:**

this works achieves adversarial robustness without adversarial training adding a "refinement" phase to a network trained with an alignment loss. The idea of the refinement is to force the model to be robust to perturbations outside of the detected item and is done using curriculum learning using a threshold on saliencies as a mask. Avoiding iterative training allows to significantly lower the computational cost for Sota robustness.
I think the discussion with reviewer Wty9 highlights well the situation: yet the paper can be considered as an empirical better way to force a NN to focus on the actually important part of an image, empirical evidence is convincing and an ablation study is performed. I'd love a discussion about the existence or not of a difference wrt the robustness obtained with iterative methods.
Also, adding results on imageNet would improve the paper and mentioning the low variance in the legends of the tables rather than in text would be good.